# Impact of Fast Urbanization on Ecosystem Health in Mountainous Regions of Southwest China

**DOI:** 10.3390/ijerph17030826

**Published:** 2020-01-28

**Authors:** Yi Xiao, Luo Guo, Weiguo Sang

**Affiliations:** College of Life and Environment Science, Minzu University of China, Beijing 100081, China; CLESXiaoYi@muc.edu.cn (Y.X.); guoluo@muc.edu.cn (L.G.)

**Keywords:** urbanization, spatial correlation, ecosystem health, comprehensive indicators, GIS

## Abstract

Accelerated urbanization has changed land use patterns, leading to the deterioration of ecosystems. Assessments of ecosystem health (ESH) during the urbanization process are used to determine the reasons and mechanism for this, and to uncover negative factors. In this study, we assessed the ESH of Qiannan prefecture, in Guizhou Province, China, based on the ecosystem services value. We selected a series of indicators, including natural, social, and economic aspects, to detect the impact of urbanization on ecosystem services in 1990, 1995, 2000, 2005, 2010, and 2015. The results show that ESH in Qiannan declined from 1990 to 2015, especially in the eastern and northern regions. Further, the results indicate that urbanization had a negative impact on ESH, of which the dominant factor was the proportion of construction land from 1990 to 2005. After 2005, moreover, the dominant factor was the gross domestic product. The impact of urbanization on EHS had spatial differences, however. The most significant negative impact was found in the east and north. After 2010, the western and central regions of Qiannan showed an urbanization trend in favor of ecosystem health. We recommend ecological restoration in regions with weak and relatively weak ESH levels to achieve sustainable development.

## 1. Introduction

Urbanization, along with economic development, population increase, and land use change, is the main path to social development [1,2,3]. Nowadays, about 55% of the world’s population lives in cities and the global urbanization rate is expected to reach 68% by 2050. Urban expansion can cause great changes in ecosystem structure and function [4,5,6]. During the process of urbanization, the flow of material, energy, and information is influenced, and the structure and function of ecosystems are affected [7]. For example, urbanization has caused increasing land fragmentation and land use pattern conversion from original to designed land cover, which has a significant influence on ecosystems, including the loss of habitats, decreases in agricultural and forest productivity, as well as the reduction of climate regulation function supply by plants [8,9]. Considering both the structures and functions of ecosystems, Rapport defined ecosystem heath (ESH) as the ability to maintain ecosystem structures and recover with self-regulating processes after disturbances [10]. This concept of ESH enriches the study of ecosystems and is widely accepted by other scientists. Many researchers have been focusing on the relationship of urbanization and ESH. Cheng [11] assessed river ecosystem health of the Haihe River Basin based on multiple indices and found that arable land, urban land, gross domestic product (GDP) per capita, and population density had negative relationships with river ecosystem health. Pan [12] explored the variation of ecosystem health level of Huzhou City under the background of urbanization and concluded that the health level deteriorated severely from 2001 to 2006. Styers [13] selected landscape indicators and urbanization variables to measure ESH. Five indicators were integrated by Niekerk to examine key pressures from urbanization and ESH [14]. Population growth and the development of urban areas require more benefits (e.g., food, clean air, and water) from ecosystems for supporting human beings [15]. Maintaining ecosystem health at a high level is important for supplying adequate resources and achieving sustainable development. For managers and decision makers, monitoring the health level of ecosystems and analyzing the effects of urbanization on ecosystems is essential for land use planning and scientific eco-environmental policy making.

Although many studies have been conducted, there is still a challenge to quantify the relationships between urbanization and ESH because it is difficult to assess ESH due to ecosystem complexity. Among different methods, the framework of vigor, organization, and resilience has been widely utilized by ecologists to assess ESH [16,17]. Based on Rapport’s theory, a healthy ecosystem can be quantified from three aspects: Vigor, organization, and resilience [18]. Vigor represents the functioning of ecosystems, which can be measured by the metabolism or primary productivity; organization represents the quantity and diversity of relationships between the components of the system, which are usually measured by landscape pattern index; resilience refers to the ability of the system to retain its structure and function under stress, which is usually measured based on land use type [19]. Based on the vigor–organization–resilience framework, He [20] assessed the ecosystem health level in China from 2000 to 2015 and explored the socio-economic factors driving the regional differences of ecosystem health. Kang used the vigor–organization–resilience framework to investigate the way in which urbanization influences the ecosystem health of the Beijing–Tianjin–Hebei urban agglomeration [21]. Li [22] visualized five indicators of vigor aspects, four indicators of organization aspects, and four indicators of resilience aspects to assess ESH. However, the traditional vigor–organization–resilience framework focuses more on ecosystem integrity and ignores the dynamics of ecosystem function maintenance. Ecosystem services, as benefits or well-beings to society, have a close connection to the functioning of ecosystems [23]. A healthy ecosystem can supply sustainable resources to meet the needs of urban residents [24]. As such, ESH assessments that only select indicators of vigor, organization, and resilience are impoverished. Peng [25] recommended four indicators; vigor, organization, resilience, and regional ecosystem services, and considered ecosystem services and the landscape structure to analyze ESH more comprehensively. An ESH theory that incorporates vigor, organization, resilience, and ecosystem services, then, offers a method of assessing ESH that considers both natural ecosystems and human well-being [26]. This method offers comprehensive quantification and spatialization of ESH levels from these four aspects. Moreover, the method integrates land use change, landscape patterns, and socioeconomic data. This method is applied in some studies about regional ecosystem health assessment. Cui [27] involved ecosystem services in ESH assessment of Zhuhai, China. Li select 27 indicators covering vigor, organization, resilience, and ecosystem service aspects to assess Beijing and Shanghai’s urban ecosystem health from 2000 to 2011 [28]. For regional ecosystem health assessment, remote sensing data and Geography Information System (GIS) technology offer advantages when monitoring and evaluating ESH [29,30]. By processing remote sensing data, information about land use change and landscape patterns can be integrated in the assessment. Sun [31] used remote sensing and GIS technology to assess the ESH of wetlands in the Amazon and Yangtze river basins. Liao [32] assessed ESH and tested the relationship between changes in ESH and land use change in karst areas. Nowadays, few studies are conducted on ESH assessment and impacts of urbanization on ESH in mountainous areas. One of the barriers is the difficulty of obtaining data due to the complex environment in mountainous areas [33]. The development of remote sensing and GIS technology facilitate the acquisition of data [34].

In this study, we analyzed the impact of urbanization on ESH based on remote sensing data and GIS technology in Qiannan Prefecture, a mountainous area in southeast China. First, we assessed the ESH dynamics using the vigor–organization–resilience–ecosystem services framework. Second, we quantified the urbanization level. Finally, we analyzed the spatial correlation between urbanization and ESH in Qiannan Prefecture by calculating the global bivariate Moran’s *I* and local bivariate Moran’s *I*. The purposes of this study were as follows: (1) To measure and map ESH and urbanization based on land use data, population data, and economic data; (2) to quantify effects of urbanization on ESH in Qiannan Prefecture. Our study will provide a reference and suggestions for devising stable and sustainable policy for local development.

## 2. Materials and Methods

### 2.1. Study Area and Data Sources

Qiannan Prefecture is located in the southern part of Guizhou province, in southwest China. The average altitude is 997 m and the annual average temperature is 19.6 °C. The major landforms are mountain plateaus, with high terrain in the northwest and low terrain in the southeast. It has among the best-preserved karst forests in the world at the same latitude. The study area includes 12 counties, autonomous counties, and cities in Qiannan, as shown in Figure 1. There are 37 ethnic groups living in the Qiannan, including Han, Buyi, Miao, Shui, Zhuang, Dai, Maonan, and Yi.

Qiannan Prefecture has experienced a fast urbanization rate from 26.86% in 2005 to 42.01% in 2015, and this has resulted in dramatic changes to the natural landscape and environment [35]. The government of Qiannan has devised an urban development strategy for 2006–2020 [36]. To develop a comprehensive urban environmental plan for the coming decades, it is necessary to reorganize the spatial relationship between ESH and urbanization. This will provide useful information and a reference for urbanization planning and ecosystem management. 

In this study, satellite images of Qiannan taken in 1990, 1995, 2000, 2005, 2010, and 2015 [37] were pre-processed with radiation calibration, geometric correction, and image clipping. The Normalized Difference Vegetation Index (NDVI) was obtained from these images. Ecosystems were classified into five types: Forest, grass, farmland, water, and desert. To ensure a reliable classification, kappa values (> 0.85) were used to evaluate the accuracy. Population and gross domestic product (GDP) data (500 m × 500 m) were supplied by the Resource and Environment Data Cloud Platform, Institute of Geographic Sciences and Natural Resources Research, and the China Academy of Science (http://www.resdc.cn/) [38]. Development plans and relevant policy bulletins were obtained from the official website of the People’s Government of Guizhou Province (http://www.guizhou.gov.cn/) [36].

### 2.2. Assessment of Ecosystem Health

In this paper, the ESH of Qiannan was assessed from two dimensions according to Peng’s method [26]: The physical health of the ecosystem, consisting of the three traditional ESH assessment indicators (vigor, organization, and resilience) [39,40]; and the ecosystem services value. A high ecosystem services value means that sufficient material and energy are delivered to society and the ecosystem is in a stable and healthy state [41]. Thus, we assumed that ecosystem services and the ESH level are positively related. The formula used is as follows [42]: (1)$$H=\sqrt{PH\times ESV}$$
where H represents the ESH level of the study area. In this study, five levels of ESH were divided: Dtrong ESH (80–100%), relatively strong ESH (60–80%), ordinary ESH (40–60%), relatively weak ESH (20–40%), and weak ESH (0–20%). PH represents the physical health of the ecosystem; and ESV represents the ecosystem services value.

For the physical health of the ecosystem, we use a formula to integrate these factors:(2)$$PH=\sqrt[{3}]{V\times O\times R}$$
where V represents ecosystem vigor; O represents ecosystem organization; and R represents ecosystem resilience. 

Exhaustively, ecosystem vigor is defined as the ecosystem’s activity, metabolism, or primary productivity [19]. In this study, the NDVI was applied to quantify ecosystem vigor, because it represents the degree of vegetation growth and primary productivity [43,44,45]. Ecosystem organization represents the number and diversity of relationships between the components [45]. In this study, ecosystem organization was assessed in terms of landscape heterogeneity and landscape connectivity. Landscape heterogeneity was quantified using Shannon’s Diversity Index (SHDI), Patch Density (PD), and the Area-Weighted Patch Fractal Dimension (AWMPFD) [46]. Landscape connectivity was measured using the Patch Cohesion Index (COHESION), Connectance Index (CONNECT), Integral Index of Connectivity (IIC), and Contagion Index (CONTAG) (Table 1).

Ecosystem resilience is the ability of an ecosystem to retain its structure and function steadily under stress [47]. Land use contributes in different ways to ecosystem resilience [48,49], and we obtain the ecosystem resilience for whole study area using the following formula:(3)$$R=\sum_{i=1}^{n}A_{i}\times R_{i}$$
where R represents ecosystem resilience; $A_{i}$ represents the area of land use type *i*; and $R_{i}$ represents the ecosystem resilience coefficient of land use type *i* (Table 2).

We obtain $R_{i}$ from researches of Guizhou Province and its adjacent areas [50,51].

### 2.3. Quantifying ESV

Based on the evaluation model proposed by Costanza [52], Xie presented the values per unit area of ecosystem services in China, which is widely used for quantifying the ecosystem services value (ESV) [53]. We obtained the ESV by using Xie’s table (Table 3) for 1990, 1995, 2000, 2005, 2010, and 2015. The ESV is calculated as follows:(4)$$ESV=\sum_{i=1}^{n}A_{i}\times P_{i}$$
where ESV represents the total ESV per unit; $A_{i}$ represents the area of land use type *i*; and $P_{i}$ represents the values per unit area of land use type *i*.

### 2.4. Mapping Urbanization Levels

To measure urbanization, there are many different variables used [54]. Summarizing them, urbanization is usually measured in three dimensions: Population increase, economic increase, and construction land expansion [55]. Population growth is a remarkable characteristic of urbanization. According to research, the proportion of the populations living in urban areas has increased to over 50% [56]. Population growth has promoted the need for more construction land for residents and has, thus, inspired economic development. Economic increase and construction land expansion have a positive correlation with urbanization. Peng [57] selected GDP density and construction land proportion to map urbanization levels. In this study, the population density (person km^−2^), GDP density (10^4^ yuan km^−2^), and construction land proportion (CLP) were selected to measure the population urbanization level, economic urbanization level, and land urbanization level, respectively.

### 2.5. Spatial Correlation Measurement

Moran’s *I* is an index of spatial correlation [58], reflecting the similarity of the spatially adjacent regions. It is widely used in research on ecological security [59], ecological vulnerability assessments [60], and ecosystem services change [61]. In this study, Moran’s *I* was utilized to study the spatial correlation between ESH and urbanization, including the global bivariate Moran’s *I* and local bivariate Moran’s *I* (bivariate Local Indicators of Partial Association (LISA)). The global bivariate Moran’s *I* was used to explore the spatial correlation between ESH and urbanization in the study area, and the local bivariate Moran’s *I* was used for spatial correlations within different spatial units [62,63]:(5)$$I=\frac{N\sum_{i}^{N}\sum_{j\neq i}^{N}W_{ij}Z_{i}Z_{j}}{\left(N-1\right)\sum_{i}^{N}\sum_{j\neq i}^{N}W_{ij}}$$
(6)$$I_{kl}^{i}=Z_{k}^{i}\sum_{j=1}^{N}W_{ij}Z_{l}^{j}$$
(7)$$Z=\frac{X_{k}^{i}-{\bar{X}}_{k}}{\sigma_{k}}$$
(8)$$Z_{l}^{j}=\frac{X_{l}^{j}-{\bar{X}}_{l}}{\sigma_{l}}$$
where I is the global bivariate Moran’s *I* for EHS and the urbanization level; $I_{kl}^{i}$ is the local bivariate Moran’s *I* for EHS and the urbanization level; N represents the total number of spatial units; $W_{ij}$ is a spatial weight matrix for measuring the spatial correlation between the *i* and *j* spatial unit [58]; $Z_{i}$ is the deviation between the attribute of the *i* spatial unit and the average of the attribute; $Z_{j}$ is the deviation between the attribute of the *j* spatial unit and the average of the attribute; $X_{k}^{i}$ is the value of attribute *k* of spatial unit *i*; ${\bar{X}}_{k}$ is the average of attribute *k*; $\sigma_{k}$ is the variance of attribute *k*. $X_{l}^{j}$ is the value of attribute *l* of spatial unit *j*; ${\bar{X}}_{l}$ is the average of attribute *l*; and $\sigma_{l}$ is the variance of attribute *l*.

The value of I/$I_{kl}^{i}$ ranges from −1 to 1. A positive I/$I_{kl}^{i}$ value indicates a positive spatial correlation between ESH and urbanization, which signifies that a unit with a high ESH level (more than the mean) [62,63] is surrounded by units with high urbanization levels. Conversely, a negative I/$I_{kl}^{i}$ indicates a negative spatial correlation, such that a unit with a high ESH level is surrounded by units with a low urbanization level (less than the mean) [62,63]. The high absolute value of I/$I_{kl}^{i}$ indicates that the spatial correlation is strong. In this study, we used permutation tests (999 permutations) to evaluate the statistical significance of the bivariate Moran’s *I* [62,63]. In order to get credible results, we set the statistical significance value at the 1% level for the spatial correlation between ESH and urbanization.

## 3. Results

### 3.1. Assessment of Ecosystem Health

#### 3.1.1. Dynamics of ESH in Qiannan

From 1990 to 2015, obvious changes were found in the total proportion of areas with relatively weak ESH, ordinary ESH, relatively strong ESH, and strong ESH. Relatively weak ESH increased from 2.02% to 13.55%, ordinary ESH increased from 19.50% to 30.03%, strong ESH increased from 21.3% to 51.73%, relatively strong ESH decreased from 55.90% to 3.36%, and weak ESH rarely changed (Figure 2).

From 1990–2005, weak ESH, relatively weak ESH, and strong ESH rarely changed. During this period, relatively strong ESH decreased sharply from 21.31% to 1.80%, while ordinary ESH increased from 19.50% to 39.83%. From 2005 to 2015, significant changes occurred in relatively weak ESH, relatively strong ESH, and strong ESH. Relatively strong ESH increased obviously from 1.80% to 51.73 % and relatively weak ESH increased from 2.01% to 13.55%. During this period, strong ESH showed a sharp decrease from 55.09% to 3.36%. Ordinary ESH decreased from 39.83% to 30.03%. The results showed that the main part of ecosystems in Qiannan had strong ESH from 1990 to 2005 and part of the area with relatively strong ESH was converted to area with ordinary ESH. Changes of five ESH levels in different areas showed that most parts of the ecosystem in Qiannan were healthy from 1990 to 2005, and deteriorating from 2005 to 2015. The decrease in ESH in Qiannan occurred in 2005 and 2015. These results indicated that ESH in Qiannan was deteriorative.

#### 3.1.2. Dynamics of ESH Spatial Patterns

The spatial distribution of ESH in Qiannan ethnic districts from 1990 to 2015 is shown in Figure 3. It can be seen that, in 1990, areas with weak ESH were mainly distributed in Wenan, Duyun, Fuquan, and Longli, the main urban areas of Qiannan ethnic districts. Areas with relatively weak ESH were mainly distributed in Sandu. Areas with ordinary ESH were located in various areas of Qiannan ethnic districts and mainly distributed around areas with weak ESH. Areas with relatively strong ESH were mainly distributed in boundary regions: Guiding, Longli, and Huishui. Areas with strong ESH were distributed in Changshun, Luodian, Sandu, and parts of Pingtang.

In the eastern part of Qiannan’s ethnic districts, ESH was strong. These areas are distributed in the Duliujiang Wetland Nature Reserve, so the environment in these areas was well protected by the government of Qiannan.

According to the results, there was a decrease in ESH in areas of the northern and eastern parts of Qiannan from 1990–2005. Areas with ordinary ESH expanded, especially in the middle and eastern parts of Qiannan. The areas with relatively strong ESH reduced, and dropped to the category of ordinary ESH. The areas with strong ESH remained stable. For example, Fuquan and Guiding experienced a decline in ESH levels in this period, and eastern Huishui was relegated to weak ESH.

In 2010, the most characteristic result was the area with strong ESH, which drastically decreased, especially in the north and east. For instance, in northern Qiannan, areas with strong ESH in Wengan, Fuquan, Longli, and Guiding showed an obvious decrease, as did Sandu in the east.

Compared to other years, 2015 had the most areas with relatively weak ESH and the least areas with strong ESH. ESH in northern Qiannan, including Wengan, Longli, and Guiding, deteriorated from ordinary ESH to relatively weak ESH, and this also occurred in Duyun (eastern Qianan) and Dushan (southern Qiannan). ESH in western Qiannan deteriorated from strong ESH to relatively strong ESH. Overall, by 2015, only 55.09% of areas in Qiannan’s ethnic districts had strong or relatively strong ESH. 

### 3.2. Dynamics of Urbanization Spatial Patterns

To verify the relationship between rapid urbanization and ESH, we characterized the spatial patterns of urbanization levels in Qiannan’s ethnic districts during the study period (Figure 4). We found that spatial patterns of economic urbanization, population urbanization, and land urbanization had similar trends. Urbanization was highest in cities, gradually decreasing from the main city area to the peripheral areas. In detail, economic urbanization was the highest in the middle part of Qiannan, especially in the main city area, considerably increasing from 1990–2015. By contrast, economic urbanization increased gradually in western Qiannan.

Land urbanization was highest in the main city, gradually decreasing to the surrounding areas. The same trend was seen with regard to economic urbanization. From 1990–2005, land urbanization increased slowly; after 2005, it expanded much faster. Areas with high land urbanization were found in the middle and northern parts of Qiannan, especially in Duyun, Guiding, Wenan, and Longli.

There was a persistent increase in population urbanization from 1990 to 2015, especially in the main city. The results showed an obvious increase from 2005 to 2015. Regions with a high level of population urbanization were found in Duyun, Guiding, and Wenan.

### 3.3. Effect of Urbanization on Ecosystem Health

The results provided by the global bivariate Moran’s *I* showed obvious negative spatial correlations between ESH and urbanization (all Moran’s *I* values < 0 and *p*-values = 0.01) (Table 4). This means that the three kinds of urbanization exerted a negative impact on ESH. Additionally, urbanization exerted various negative pressures in different years.

The negative correlation between land urbanization and ESH, and between economic urbanization and ESH, increased from 1990–2015. In 1990, there was a strong negative correlation between land urbanization and ESH (Moran’s *I*: −0.072), followed by that between economic urbanization and ESH (Moran’s *I*: −0.068), and population urbanization and ESH (Moran’s *I*: −0.052). In 2000, the negative correlation between the three kinds of urbanization and ESH showed a trend similar to that in 1990. The Moran’s *I* between land urbanization, economic urbanization, and population urbanization was −0.074, −0.068, and −0.035, respectively, but from 2005 to 2015, the negative correlation was strongest between economic urbanization and ESH, followed by that between land urbanization and ESH, and between population urbanization and ESH.

The bivariate LISA results showed four types of spatial correlations between urbanization and ESH (Figure 5). The effects of population increase, GDP growth, and construction land expansion on ESH of Qiannan Prefecture varied. The similarity was reflected in the expansion of high ESH and low urbanization (HL) areas which are negatively affected by three indicators in 1990–2015. 

HL areas for GDP and ESH were mainly concentrated in the north and middle parts of the areas around the main city in Qiannan ethnic districts in 1990. With the development of the urban economy, the areas of HL increased significantly, especially from 2005–2015. HL areas for population density (POP) and ESH, and for construction land proportion (CLP) and ESH, increased in the period of 2005–2015. In addition, areas of HL were concentrated in the area surrounding the main city. 

Low ESH and high urbanization (LH) areas for the three types of spatial correlations all decreased, along with urbanization in the period from 2005 to 2015, with a peak in 2015. LH areas for POP and GDP were distributed in the western and eastern parts of Qiannan. LH areas for CLP were distributed in the western, eastern, and northern parts of Qiannan.

High ESH and high urbanization (HH) areas for GDP and ESH appeared in the middle of Qiannan in 2015, where Pingtang National Geological Park is located. After 2010, tourism income from this park increased. HH regions for POP and ESH appeared in the western and middle parts of Qiannan in 2015, where traditional village tourism regions are distributed, attracting many tourists.

Three types of low ESH and low urbanization (LL) areas appeared in eastern part of Qiannan from 2010 to 2015, an area with stony desertification. The three types of LL areas tended to expand.

## 4. Discussion

### 4.1. Change in ESH in Qiannan from 1990 to 2015

We evaluated the spatial distribution of ESH over six years. The results showed that the areas with relatively weak ESH were mainly located in karst areas, which erode easily and mainly comprise limestone. This kind of area constantly increased from 1990 to 2015. If this trend continues, the vigor and resilience of the ecosystem will be compromised. The ecosystem will degrade and stony desertification in the region will intensify. Areas with strong ESH were located widely throughout Qiannan in 1990 but deteriorated to relatively strong ESH thereafter, and they were mainly found in western parts of Qiannan in 2015. The stable levels in western Qiannan were attributed to slower economic development and less disturbance. Another reason for this is the so-called Green-for-Grain Project, which was implemented in western Qiannan [64].

This study detected the spatial relationship between ESH and urbanization, for a more complete understanding of the impact of urbanization on ESH. The results showed a strong connection between urbanization and ESH. In this study, decreasing ESH in some regions resulted from an increase in urbanization (all bivariate Moran’s *I* < 0; Table 4). The results of Moran’s *I* Index showed that GDP growth had the greatest negative impact on ESH after 2005. Massive land exploitation activities occurred in Qiannan, with 163 land exploitation programs conducted from 2006 to 2014 [37]. These activities were concentrated in the northern and eastern regions, promoting economic development but destroying forests. The structure and function of ecosystems were damaged, leading to a decline in ESH. The bivariate LISA (Figure 5) and significant correlation of the urbanization indicators (Table 4) showed that urbanization exerted a negative impact on ESH in Qiannan. This indicated that urbanization contributed to the variation of ESH at a regional scale. On one hand, an increase in construction land for urbanization occupies a large amount of land resources, changes the physical properties of the soil, and reduces drainage, with less heat absorption and heat dissipation, ultimately affecting the function of the soil system [65]. On the other hand, the impact of increased construction land in urban areas on the ecosystem is mainly due to the increase in population density. The discharge of garbage and sewage tends to make the ecosystem unbalanced, as species decrease and the functioning of the ecosystem declines [66]. If the declining trend continues in the future, the ESH of Qiannan will suffer serious deterioration. Ultimately, the ecosystem may not have the ability to support sustainable development. We think that a land use planning policy that includes continual ESH assessments and monitoring is necessary for decision makers to consider. This is neglected in current planning.

### 4.2. Spatial Relationship between ESH and Urbanization for Ecosystem Management

Urban planners and environmental managers have always faced the question of how to balance the relationship among urban construction, economic development, and ecological conservation. Although some solutions are available, such as demarcating ecological protection areas [67,68], many problems remain. For instance, studies neglect to consider the impact of urbanization on ecosystem conservation. Construction activities (roads, artificial pond, reservoirs, etc.) in the background of high economic output can fragment the land, negatively impacting the ecosystem services supply [69]. Based on this concern, we offer several recommendations for better and environmentally-friendly urban planning based on our results.

ESH assessments can provide useful information about the state of an ecosystem. It is important to conserve ecosystems and devise sound policy for regions under rapid urbanization. We suggest increasing the so-called Green GDP [70] as a target for achieving sustainable development in ethnic districts, rather than merely considering economic growth. We also suggest that the ecosystem services value should be included in GDP accounting in Qiannan, because it reflects the links between urbanization and ESH to some extent. In this way, the interactions between humans and the ecosystem can be explored and institutionalized. Relevant data can be a helpful reference for a more scientific development policy in Qiannan.

After 2010, national parks in Qiannan achieved high GDP while maintaining strong ESH. National parks in Qiannan are part of the Protected Area System of China [71]. The Protected Area System of China has achieved progress in ecosystem protection and economic development, and it provides a new method of accomplishing sustainable development in Qiannan.

Our study identified spatial and temporal relationships between three indicators and ESH at a local scale. We believe that zoning and management is crucial for Qiannan. For high ESH and high urbanization regions (Figure 5), intensive land exploitation should be restricted. Land use types that are important for supplying ecosystem services and maintaining a healthy ecosystem should be conserved for sustainable development. In regions with strong ESH and low urbanization, it is necessary to implement a rigorous policy to protect the ecosystem against human disturbance. In regions with low ESH and high urbanization, the ecosystem is delicate and easily disturbed by humans. Thus, artificial green infrastructure is needed to improve the ecosystem services supply. For regions with low ESH and low urbanization, only moderate land activity should be allowed. In these regions, we hope tourism and ecosystem conservation can promote each other. For example, ecotourism attractions can be established by combining traditional ethnic cultural tourism and ecosystem conservation projects [72]. 

## 5. Conclusions

We concluded that; (1) the ESH in Qiannan declined from 1990 to 2015, with a more obvious decline in eastern and northern regions than that in western regions; (2) rapid urbanization had the greatest negative impact on ESH in eastern and northern regions. The dominant factor from 1995 to 2005 was CLP, and after 2005 GDP was the dominant factor; and (3) the western and central regions of Qiannan showed an urbanization trend in favor of ESH after 2010. We recommend ecological restoration in the regions with poor ESH, and ecotourism should be encouraged to achieve sustainable development.

## Figures and Tables

**Figure 1 ijerph-17-00826-f001:**
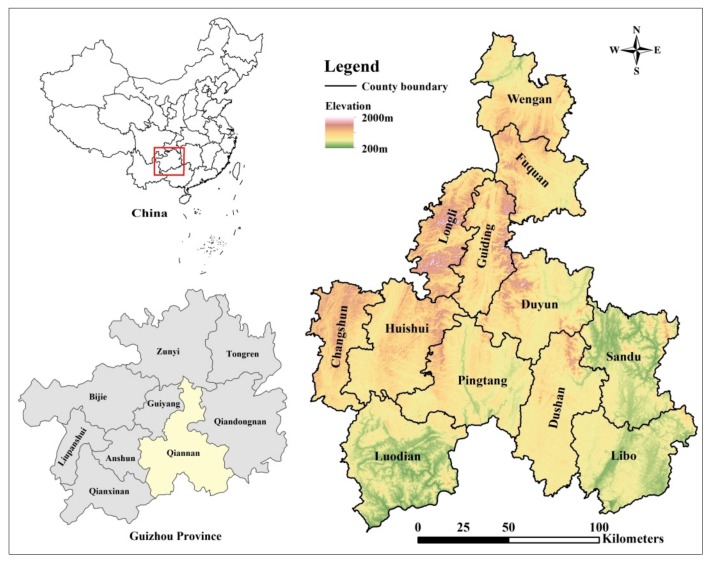
The studying location of Qiannan Prefecture in North of Guozhong Province and China (red box area).

**Figure 2 ijerph-17-00826-f002:**
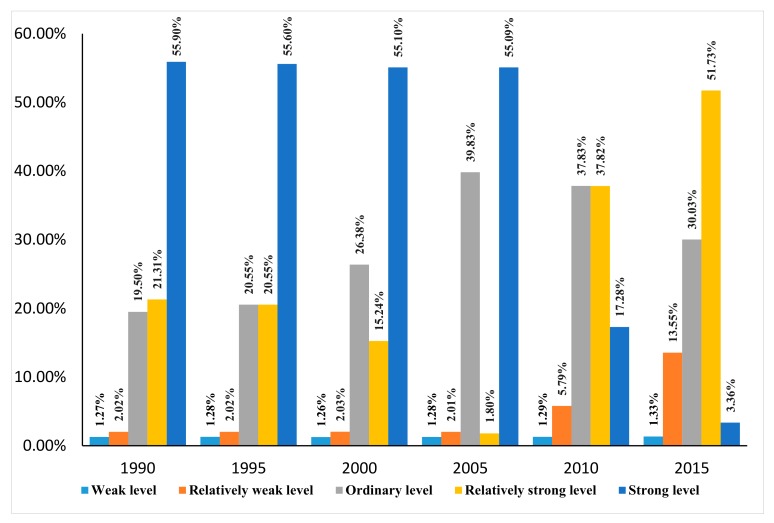
Proportion of areas with different ecosystem health (ESH) levels from 1990 to 2015.

**Figure 3 ijerph-17-00826-f003:**
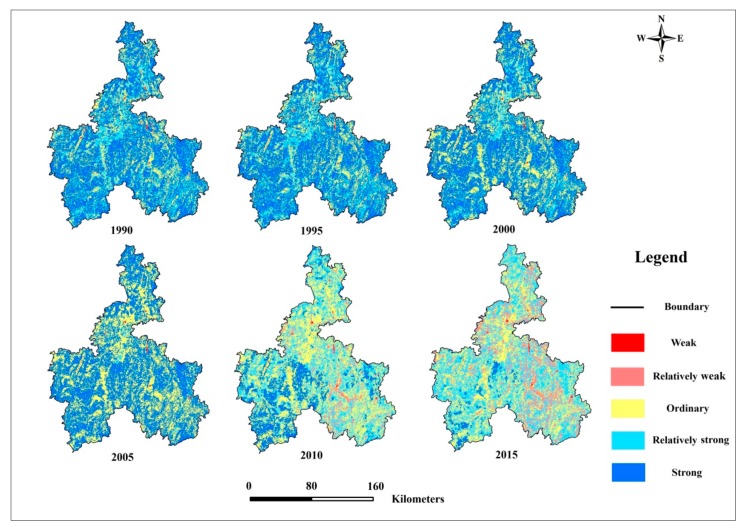
Spatial patterns of ESH from 1990 to 2015.

**Figure 4 ijerph-17-00826-f004:**
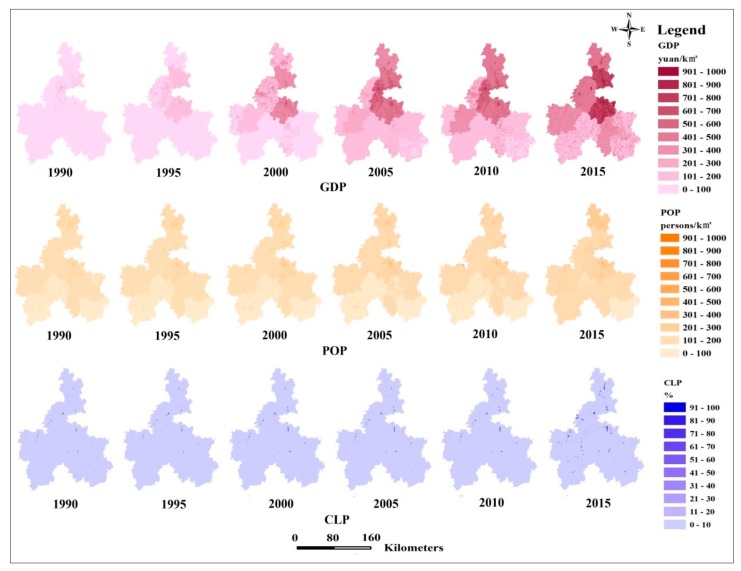
Dynamics of spatial patterns of urbanization in Qiannan’s ethnic districts (GDP: GDP density; CLP: Construction land proportion; POP: Population density).

**Figure 5 ijerph-17-00826-f005:**
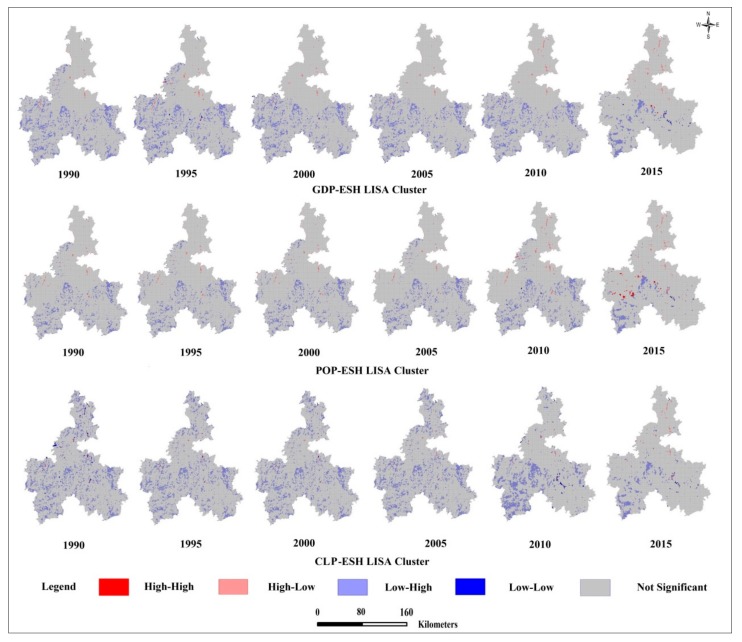
LISA (local indicators of spatial association) cluster maps between ESH and urbanization level. (GDP: GDP urbanization; CLP: Construction land proportion; POP: Population urbanization; HH: High ESH and high urbanization; HL: High ESH and low urbanization; LH: Low ESH and high urbanization; LL: Low ESH and low urbanization).

**Table 1 ijerph-17-00826-t001:** Weight of factors measuring organization.

Indicator	Factor		Weight
Organization	PD	Patch Density	0.2
	SHDI	Shannon’s Diversity Index	0.3
	AWMPFD	Area-Weighted Patch Fractal Dimension	0.1
	COHESION	Patch Cohesion Index	0.1
	CONTAG	Contagion Index	0.1
	CONNECT	Connectance Index	0.1
	IIC	Integral Index of Connectivity	0.1

Notes: PD- Patch Density in landscape; SHDI- Shannon’s Diversity Index in landscape; AWMPFD-he Area-Weighted Patch Fractal Dimension in landscape; CONNECT- Connectance Index in landscape; CONTAG- Contagion Index in landscape; IIC- Integral Index of Connectivity in landscape.

**Table 2 ijerph-17-00826-t002:** Ecosystem resilience coefficient of land use type [52,53].

Ecosystem Type	Forest	Grass	Water	Farmland	Desert
*R*	0.9	0.8	0.8	0.5	0.1

**Table 3 ijerph-17-00826-t003:** Values per unit area of ecosystem services in China [53] Unit: yuan hm ^−2^ yr ^−1^.

Ecosystem Services		Forest	Grassland	Water	Farmland	Desert
Provisioning service	Food production	148.20	193.11	449.10	238.02	8.98
	Raw materials	1338.32	161.68	175.15	157.19	17.96
Regulating service	Gas regulation	1940.11	673.65	323.35	229.04	26.95
	Climate regulation	1827.84	700.60	435.63	925.15	58.38
	Water regulation	1836.82	682.63	345.81	8429.61	31.44
	Waste treatment	772.45	592.81	624.25	6669.14	116.77
Supporting service	Soil formation & protection	1805.38	1,005.98	660.18	184.13	76.35
	Biodiversity maintenance	2025.44	839.82	458.08	1540.41	179.64
Cultural service	Recreation & aesthetic value	934.13	390.72	76.35	1,994.00	107.78
	total	12,628.69	5241.00	3547.89	20,366.69	624.25

**Table 4 ijerph-17-00826-t004:** Bivariate Moran’s *I* between ESH and economic urbanization, population urbanization, and land urbanization.

Factors	Year	1990	1995	2000	2005	2010	2015
GDP ^1^	Moran’s *I*	−0.068	−0.071	−0.068	−0.077	−0.13	−0.11
	*z*-value	−72.56	−71.74	−63.40	−72.82	−116.36	−97.56
	*p*-value	0.01	0.01	0.01	0.01	0.01	0.01
POP ^1^	Moran’s *I*	−0.052	−0.042	−0.035	−0.036	−0.082	−0.067
	*z*-value	−48.21	−37.22	−32.40	−35.26	−80.56	−67.35
	*p*-value	0.01	0.01	0.01	0.01	0.01	0.01
CLP ^1^	Moran’s *I*	−0.072	−0.074	−0.074	−0.076	−0.083	−0.085
	*z*-value	−83.20	−76.34	−73.22	−66.13	−74.67	−77.29
	*p*-value	0.01	0.01	0.01	0.01	0.01	0.01

^1^ Statistical significance at the 1% level.

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
