# Peer review of "Impact of Fast Urbanization on Ecosystem Health in Mountainous Regions of Southwest China"

_ijerph, 2020, doi:10.3390/ijerph17030826_

Round 1

Reviewer 1 Report

This paper analyzes the ecological health of Qiannan prefecture from 1990 to 2015 and its relationship with urbanization. It has a good reference significance for understanding the change of ecological health and the impact of urbanization, improving ecological environment, and coordinating ecological protection and urban development in Qiannan prefecture.

In Introduction, the first paragraph should be appropriately expanded, such as adding some data to enrich the research background, reflecting the current research needs of ecological health, and its correlation with urban development and people's living standard, either on large scale or small scale. In the second paragraph, the research meaning of ESH should be related to the "analyzing the effects of urbanization" mentioned in the first paragraph. In view of the concern for ecosystems in recent years, the author should appropriately expand the research review in the third paragraph. And, while the fourth and fifth paragraphs describe the deficiencies of existing research, they are not clear enough. Many improvements in existing research are proposed, but still need to consider how to reflect the shortcomings in more detail. They should not be simply listed and introduced, but more importantly, to contact with this article to reflect its inheritance, supplement, and development. The "remote sensing data and GIS technology" in the fifth paragraph is the current research and technology improvement. Please consider whether these should be separated into a paragraph or merged with the fourth paragraph. For the last paragraph, the meaning and contribution of this paper, an introduction to each paragraph, and an overview of the methods should be added. The content in first paragraph of 2. Assessment of Ecosystem Health has been detailed in Introduction and need not be mentioned again. The "where ?? represents the physical health of the ecosystem" in line 110 is consistent with the meaning of PH in line 107 and can be omitted. The "ecosystem resilience coefficient" set in Table 2 on line 127 should indicate the source or reason. The content in 4. Mapping urbanization levels did not explain the source or reason for the selection of the four indicators describing the level of urbanization, and the characterization of "living quality amelioration" was not clear. For the four "spatial correlations" of "bivariate LISA results" in 3. Effect of urbanization on ecosystem health, some differences should be added, as well as a simple explanation of similarities and differences. Some reasonable explanations for "Moran's I values" should be added in 1. Change in ESH in Qiannan from 1990 to 2015 to explain the different effects of urbanization indicators on ecological health.

Author Response

In Introduction, the first paragraph should be appropriately expanded, such as adding some data to enrich the research background, reflecting the current research needs of ecological health, and its correlation with urban development and people's living standard, either on large scale or small scale. In the second paragraph, the research meaning of ESH should be related to the "analyzing the effects of urbanization" mentioned in the first paragraph.

Response:

We rewrote the introduction and integrated the first paragraph and the second paragraph. Some data on global urbanization was added to show the general condition of urbanization in the world and current research needs (Lines 26-29). research reviews on ESH research meaning linking the urbanization and Ecosystem heath was added to show the effects of urbanization (Lines 30-47).

In view of the concern for ecosystems in recent years, the author should appropriately expand the research review in the third paragraph.

Response:

We have improved and expanded the research reviews on ESH (Lines 54-66).

And, while the fourth and fifth paragraphs describe the deficiencies of existing research, they are not clear enough. Many improvements in existing research are proposed, but still need to consider how to reflect the shortcomings in more detail. They should not be simply listed and introduced, but more importantly, to contact with this article to reflect its inheritance, supplement, and development. The "remote sensing data and GIS technology" in the fifth paragraph is the current research and technology improvement. Please consider whether these should be separated into a paragraph or merged with the fourth paragraph.

Response:

We introduced the vigor-organization-resilience-ecosystem services framework and its application (Lines 72-81), and further explained the shortcomings of the vigor-organization-resilience framework (Lines 66-70). The literature review of the application of remote sensing and GIS technology in studying effects of urbanization on regional ESH was finished and gave their inheritance, supplement and development, and the usage of remote sensing and GIS technology method in ESH researches in mountain area was analyzed (Lines 81-90).

For the last paragraph, the meaning and contribution of this paper, an introduction to each paragraph, and an overview of the methods should be added.

Response:

We have improved the last paragraph of introduction section according to reviewer’s suggestions (Lines 91-98), the meaning and contribution of this research was provided.

The content in first paragraph of 2. Assessment of Ecosystem Health has been detailed in Introduction and need not be mentioned again. The "where ?? represents the physical health of the ecosystem" in line 110 is consistent with the meaning of PH in line 107 and can be omitted. The "ecosystem resilience coefficient" set in Table 2 on line 127 should indicate the source or reason.

Response:

We deleted excess content and added the necessary sentences as reviewer’s suggestions.

The "ecosystem resilience coefficient" set in Table 2 on line 127 should indicate the source or reason.

Response:

We indicated the source of ecosystem resilience coefficient (Lines 153-154).

The content in 4. Mapping urbanization levels did not explain the source or reason for the selection of the four indicators describing the level of urbanization, and the characterization of "living quality amelioration" was not clear.

Response:

We are sorry that an error existed in our manuscript and we corrected the number of indicators describing the level of urbanization, with three indicators excluding "living quality amelioration". We also added explanation on selection of indicators describing the level of urbanization (Lines 167-176).

For the four "spatial correlations" of "bivariate LISA results" in 3. Effect of urbanization on ecosystem health, some differences should be added, as well as a simple explanation of similarities and differences.

Response:

We added explanation of similarities and differences (Lines 287-291). The spatial similarities and differences between the three urbanization indicators and ESH were illustrated (292-306) and discussed in detail (Lines 331-343).

Some reasonable explanations for "Moran's I values" should be added in 1. Change in ESH in Qiannan from 1990 to 2015 to explain the different effects of urbanization indicators on ecological health.

Response:

We added explanations for "Moran's I values"(Lines 323-331).

Reviewer 2 Report

Line 27, "impact human being" too vague. negative or positive? and people get more benefits from urbanization. Damage to natural ecosystems is more threatening to wildlife instead of human being.

Line 29, water cycle is a big concept but photosynthesis is in small scale. probably change photosynthesis to productivity. 

Line 32, ...provide an...

Line 39-41, to be consistent, add sentences describing how to measure resilience and organization.

Line 42, keep the order of three aspects consistent, so vigor, resilience and then organization.

Line 43, "...assess ESH in cities"?

Line 52-58, was the method that adding the aspect of regional ecosystem services being accepted by the society (peers)? Or it has some limitations for example, only applied to data in large scale (multiple cities)?

Line 60-64, so most of these examples are not in cities? probably authors could add some sentences emphasize the current gaps to highlight the importance of this study (using remote sensing data sets).

Line 66, extra space before "we analyzed.."

Line 67, omit "as follows"

Line 74, why the average annual temperature is a range? 

Line 76, "include.." 

Line 79, a little confused by the numbers here, so 26.86% per year? or 26.86% in 2005 to 42.01% in 2015? please clarify it.

Line 103, what is viz. ?

Line 137-138, vague. how many were used in this study? four?

Line 147-149, these three sentences are sort of repetitive, please simplify them to one sentence.

Line 175-178, belong to methods

Line 178-179, belong to Figure caption

Line 184, how much is the "little"?

Line 186-190, I suggested to calculate and put how much % of increase or decrease to the sentences in addition to the original values.

Line 202, would a pie chart instead of a bar graph be better?

Line 231, "..we characterized.." and replace "detected" with "characterized" throughout the manuscript

Line 240, "..expanded much faster"

Line 294, extra space before "mainly found..."

Line 331, is there a threshold saying above that value can be called "good ESH"?

Line 337, above what values can be called "high"? please give some specific values to make it clear. And for line 343, below what values can be called "low"?

Line 345, what actions can be called "ecotourism"? probably add a reference to help explain?

Line 348-357, some of these sentences can be combined and simplified. For example: "the ESH in Qiannan declined from 1990 to 2015, with more declines in the western regions than in the eastern and northern regions..."

Author Response

1.Line 27, "impact human being" too vague. negative or positive? and people get more benefits from urbanization. Damage to natural ecosystems is more threatening to wildlife instead of human being.

Response:

We have re-written this part, and added context about the impact of urbanization on ecosystem structure and function (Lines 29-31).

Line 29, water cycle is a big concept but photosynthesis is in small scale. probably change photosynthesis to productivity. Line 32, ...provide an...

Response:

We changed the example (Lines 31-35) and used “productivity” as suggestions from review.

Line 39-41, to be consistent, add sentences describing how to measure resilience and organization. Line 42, keep the order of three aspects consistent, so vigor, resilience and then organization.

Response:

We added sentences to describe how to measure resilience and organization (Lines 57-61). We used the order of three aspects consistent (vigor, organization and then resilience) in our whole paper.

Line 43, "...assess ESH in cities"?

Response:

We are sorry that “in cities” is an inappropriate explanation. We used “region scale” instead of “in cities”.

Line 52-58, was the method that adding the aspect of regional ecosystem services being accepted by the society (peers)? Or it has some limitations for example, only applied to data in large scale (multiple cities)?

Response:

The method has been applied in many researches on ESH recently, mainly on regional scale (Lines 79-86).

Line 60-64, so most of these examples are not in cities? probably authors could add some sentences emphasize the current gaps to highlight the importance of this study (using remote sensing data sets).

Response:

These studies are on a regional scale. Now, it is difficult to study ESH in mountain areas. We have added some sentences to highlight the importance of this study (Lines 87-90).

Line 66, extra space before "we analyzed.."

Line 67, omit "as follows"

Line 74, why the average annual temperature is a range? 

Line 76, "include.." 

Line 79, a little confused by the numbers here, so 26.86% per year? or 26.86% in 2005 to 42.01% in 2015? please clarify it.

Response:

We corrected these mistakes and unified the format of the data, please refer to the new added text.

Line 103, what is viz. ?

Line 137-138, vague. how many were used in this study? four?

Line 147-149, these three sentences are sort of repetitive, please simplify them to one sentence.

Line 175-178, belong to methods

Line 178-179, belong to Figure caption

Line 184, how much is the "little"?

Response:

We corrected the mistakes and improved the explanation by the reviewer’s suggestions and comments.

Line 186-190, I suggested to calculate and put how much % of increase or decrease to the sentences in addition to the original values.

Response:

We have re-written these 2 paragraphs according to the suggestions (Lines 204-218).

Line 202, would a pie chart instead of a bar graph be better?

Response:

Thanks for your comments. We compared the pie chart and the bar char and found that the bar chart can more intuitively show the change trend of the five ESH levels over time. Some data values are small and pie charts are more difficult to show results.

Line 231, "..we characterized.." and replace "detected" with "characterized" throughout the manuscript

Line 240, "..expanded much faster"

Line 294, extra space before "mainly found..."

Line 331, is there a threshold saying above that value can be called "good ESH"?

Response:

We corrected the mistakes and improved the explanation.

Line 337, above what values can be called "high"? please give some specific values to make it clear. And for line 343, below what values can be called "low"?

Response:

  We added the meaning of "high" and "low" in the methods part (Line 192, Line 195).

Line 345, what actions can be called "ecotourism"? probably add a reference to help explain?

Response:

We added references “Orams, M.B. Towards a more desirable form of ecotourism. Tour. Manag. 2012, 315–323.

Line 348-357, some of these sentences can be combined and simplified. For example: "the ESH in Qiannan declined from 1990 to 2015, with more declines in the western regions than in the eastern and northern regions..."

Response:

We simplified these sentences (Lines 380-387).

Reviewer 3 Report

This paper is very interesting concerned with ecosystem health in China and it has exellent attemptes and results!

However, if you can add, please consider below matters.

Ecosystem health including tree and plant physiology such as photosynthesis.  How haveplants been changed physiologically by fast urbanization?  If you could, please add. What does economic urbanization mean precisely?  Doe it mean making factories, consumption area, office towns? If you can describe, please add. 

Author Response

Ecosystem health including tree and plant physiology such as photosynthesis.  How have plants been changed physiologically by fast urbanization?  If you could, please add.

Response:

We just took the “photosynthesis” as an example here. We have changed “photosynthesis” to “Productivity” according to another reviewer’s suggestion (Lines 31-35).

What does economic urbanization mean precisely? Does it mean making factories, consumption area, office towns? If you can describe, please add.

Response:

It is acknowledged that economic growth is an important manifestation of urbanization. Peng (Peng, J.; Tian, L.; Liu, Y.; Zhao, M.; Hu, Y.; Wu, J. Ecosystem services response to urbanization in metropolitan areas: Thresholds identification. Sci. Total Environ. 2017, 607608, 706–714. https://doi.org/10.1016/j.scitotenv.2017.06.218

) used GDP density as indicator to measure economic urbanization (Line 171) but he didn’t explain the definition or component of economic urbanization.